# Patient safety in inpatient mental health settings: a systematic review

Bethan Thibaut,[1] Lindsay Helen Dewa [ID],[1] Sonny Christian Ramtale,[1] Danielle D'Lima,[2] Sheila Adam,[1] Hutan Ashrafian [ID],[1] Ara Darzi,[1] Stephanie Archer [ID] [1,3]

BT and LHD are joint first authors.

[1]NIHR Imperial Patient Safety Tranlsational Research Centre, Department of Surgery and Cancer, Imperial College London, London, UK
[2]Centre for Behaviour Change, Department of Clinical, Educational and Health Psychology, University College London, London, UK
[3]Department of Public Health and Primary Care, University of Cambridge, Cambridge, UK

**Correspondence to**
Dr Stephanie Archer;
stephanie.archer@imperial.ac.uk

## ABSTRACT

**Objectives** Patients in inpatient mental health settings face similar risks (eg, medication errors) to those in other areas of healthcare. In addition, some unsafe behaviours associated with serious mental health problems (eg, self-harm), and the measures taken to address these (eg, restraint), may result in further risks to patient safety. The objective of this review is to identify and synthesise the literature on patient safety within inpatient mental health settings using robust systematic methodology.

**Design** Systematic review and meta-synthesis. Embase, Cumulative Index to Nursing and Allied Health Literature, Health Management Information Consortium, MEDLINE, PsycINFO and Web of Science were systematically searched from 1999 to 2019. Search terms were related to 'mental health', 'patient safety', 'inpatient setting' and 'research'. Study quality was assessed using the Hawker checklist. Data were extracted and grouped based on study focus and outcome. Safety incidents were meta-analysed where possible using a random-effects model.

**Results** Of the 57 637 article titles and abstracts, 364 met inclusion criteria. Included publications came from 31 countries and included data from over 150 000 participants. Study quality varied and statistical heterogeneity was high. Ten research categories were identified: interpersonal violence, coercive interventions, safety culture, harm to self, safety of the physical environment, medication safety, unauthorised leave, clinical decision making, falls and infection prevention and control.

**Conclusions** Patient safety in inpatient mental health settings is under-researched in comparison to other non-mental health inpatient settings. Findings demonstrate that inpatient mental health settings pose unique challenges for patient safety, which require investment in research, policy development, and translation into clinical practice.

**PROSPERO registration number** CRD42016034057.

## Strengths and limitations of this study

► This is the first review to examine patient safety within inpatient mental health settings that uses robust systematic methodology.
► The use of a robust patient safety taxonomy provides a comprehensive list of all incident types and resulted in a wide coverage of publications in terms of setting, country and population.
► This review only included peer-reviewed studies with primary data.
► The last systematic literature search was conducted on 27 June 2019, meaning that literature published since this date will not have been included.

(eg, self-harm), and the measures taken to address these (eg, restraint), may result in further risks to patient safety.[2–6] There may also be a tension between maximising patient safety and maintaining patient autonomy. Inpatient services will often include patients who are experiencing high levels of mental distress and are therefore at greatest risk.

While mental health research has focused on components of quality of care, published research lacks focus on the science of patient safety[7–9]; the stigma and discrimination associated with mental health problems may contribute to this relative neglect.[7] Only two reviews have examined patient safety in a mental health context and described factors that influence patient safety.[7 10] These reviews highlighted the complexity of patient safety in mental health, including the importance of wider organisational safety culture. While these reviews offer important insights into this complex topic, only a small number of specific patient safety incidents and concepts were examined. As such, the current breadth and depth of patient safety research in inpatient mental health settings is unknown.

The review presented here is exploratory in nature; building on previous reviews, we aimed to report an overview of the existing research base on patient safety in inpatient

## INTRODUCTION

Patient safety has been defined as the 'avoidance, prevention and amelioration of adverse outcomes or injuries stemming from the process of healthcare'.[1] Those receiving care in inpatient mental health settings face similar risks (eg, medication errors) to patients in other areas of healthcare. In addition, some of the unsafe behaviours associated with serious mental health problems

mental health settings. We also aimed to critically reflect on quality and methods used in included studies in the field.[11] In addition to our original protocol,[11] we aimed to collate, describe and construct the main research categories, allowing for an easily accessible reference index.

## SEARCH STRATEGY AND SELECTION CRITERIA

A systematic search was developed in line with the Preferred Reporting Items for Systematic Reviews and Meta-Analyses (PRISMA) guidelines.[12] The protocol for this systematic review has been published elsewhere.[11]

Six databases were searched: Embase, Cumulative Index to Nursing and Allied Health Literature (CINAHL), Health Management Information Consortium (HMIC), MEDLINE, PsycINFO and Web of Science. The search was originally conducted on 5 April 2016 and then updated on 27 June 2019 using a comprehensive list of search terms (n=343) related to 'mental health' (n=73), 'patient safety' (n=206), 'inpatient setting' (n=13) and 'research' (n=51); see online supplementary files 1 and 2 for full search criteria and terms. The search terms included in the 'patient safety' facet were based on the National Reporting and Learning System (NRLS) taxonomy for England and Wales[13] to ensure all incident types were identified in the search. A Google Scholar search using the main search terms was also conducted; it was originally anticipated that the first 20 pages of Google scholar would need to be screened against criteria,[11] but screening stopped at five pages as no new publications were retrieved. Similarly, we had anticipated hand-searching references of all included papers within the review. However, due to the large number of papers included in the review, only the reference lists of the two existing systematic reviews were searched for additional references.

Five reviewers (BT, CR, LD, DD and SAr) screened all titles against the inclusion and exclusion criteria, with 10% independently screened by a second reviewer (split equally between BT, CR, LD, DD and SAr). Full definitions and descriptions of these criteria can be found in online supplementary file 1 and the protocol published elsewhere.[11] Inclusion and exclusion criteria were developed over several iterative rounds among the research team to ensure consistency between reviewers (online supplementary file 1). Any disagreements between reviewers were resolved through discussion and an overall consensus was obtained. Agreement between reviewers was calculated using Cohen's kappa,[14] which is a widely accepted measure of inter-rater reliability.[15 16] Full-text papers were assessed for inclusion by two reviewers from the research team (BT and one other from CR, LD and SAr); a third reviewer (DD) was consulted if necessary.

Inclusion criteria:
▶ Population: mental health inpatients;
▶ Intervention/outcomes: patient safety outcomes;
▶ Setting: inpatient setting;
▶ Comparators: no restriction;

▶ General inclusion criteria: empirical peer-reviewed studies with a clear aim or research question, that used primary data and written up in the English language between 1 January 1999 and 27 June 2019 (in line with the publication of the Institute of Medicine's report 'To Err is Human: Building a Safer Health System').[17]

Exclusion criteria:
▶ Population: centres on physical healthcare patients;
▶ Intervention/outcomes: patient safety was not the central aim, research question or outcome
▶ Setting: amalgamation of data from inpatient and outpatient settings (where inpatient sample cannot be separated out); primary care, outpatient mental health services, community or social care settings and risk assessment tool reliability/validity checks;
▶ Comparators: no restrictions;
▶ General exclusion criteria: secondary data, not in English language, protocols, editorials, commentaries/clinical case reviews/'snapshot' studies of a patient group, book chapters, conference abstracts, audits, dissertations, epidemiological studies and reviews.

## QUALITY ASSESSMENT

Quality assessment was performed to give an overview of the methodological rigour of included studies and to support readers' interpretation of the literature. Publications were not excluded based on poor quality because the review was purposively exploratory and all-encompassing. Quality was assessed by four reviewers (BT, CR, LD and SAr) using the tool derived by Hawker et al,[18] to allow appropriate assessment of the wide variety of studies included in this review. The checklist by Hawker et al evaluates nine domains: 1) abstract/title; 2) introduction and aims; 3) method and data; 4) sampling; 5) data analysis; 6) ethics and bias; 7) results; 8) transferability and generalisability and 9) implications and usefulness. For each study, the nine domains were assessed using one of four quality categories: very poor (10 points), poor (20 points), fair (30 points) and good (40 points). The scores for each study were then summed and divided by nine to get an average score.

## DATA EXTRACTION

Data were extracted by five reviewers (BT, CR, LD, DD and SAr) using a standardised form that included study design information, participant characteristics, intervention description and patient safety outcomes. Extractions were compared within the research team to ensure reliability. Only published data were extracted; study authors were contacted only for confirmation or information clarity. If the contact attempt was unsuccessful, the article was assessed in its current form.

## DATA SYNTHESIS

Studies were grouped into research categories through consensus. First, four research team members (BT, CR,

LD and SAr) individually re-read the included full-text publications and assigned each one based on the main topic area (eg, aggression). Second, each assigned topic area was checked by another team member to ensure reliability. Third, topic areas were grouped into broader research categories (eg, interpersonal violence) that best described the patient safety focus for easier navigation of the literature. Finally, these categories and the related subcategories (initially called topic areas) from the previous stage were finalised after group discussion and consensus was reached. This was to ensure mutual exclusivity and appropriate definition (table 1 and online supplementary file 3). Where data allowed, meta-analysis was performed applying a random-effects model, specifically calculating pooled prevalence considering both between-study and within-study variances that contributed to study weighting. Pooled values and 95% CIs were computed and represented on forest plots. Statistical heterogeneity was determined by the $I^2$ statistic, where <30% is low, 30%–60% is moderate and >60% is high. Analyses were performed using Stata V.15 (StataCorp, College Station, Texas, USA).

## PATIENT AND PUBLIC INVOLVEMENT
Patients and the public were not involved in this study.

## RESULTS
The search resulted in 79 672 records (figure 1) and reduced to 57 637 after de-duplication. Titles and abstracts were screened and excluded if they did not satisfy inclusion criteria (BT, CR, LD, DD and SAr). Ten per cent were then screened (n=5763) by a second independent reviewer (split equally between BT, CR, LD, DD and SAr), in line with guidance on improving decision making by including more than one person in this process[19]; good agreement was found between pairs of reviewers (κ=0.72). A total of 4758 publications were subjected to full-text review (BT, CR, LD and SAr). Two reviewers independently screened the full-text articles against inclusion criteria (BT, CR, LD and SAr). The third reviewer (DD) was consulted 59 times. Substantial agreement was reached (κ=0.64). From the full-text review, 4394 publications were excluded. Three hundred and sixty-four publications met the inclusion criteria and data were extracted (online supplementary file 4).

## Study characteristics
Table 1 provides an overview of the study characteristics . The publications spanned 5 continents and 31 countries. The three countries contributing the greatest number of studies were the UK (n=102), the USA (n=55) and Australia (n=32). The included studies collected data from over 150 000 participants. Studies included staff (n=165; 45%), patients (n=120; 33%) and a mixture of staff, patients and/or carers (n=77; 21%). Only one study focused solely on patient family members (<1%). Most studies were quantitative in nature (n=192; 53%), just over a third were qualitative (n=133; 37%) and a small proportion used mixed methodology (n=39; 11%). Studies were conducted in a variety of settings comprising: psychiatric inpatient wards/facilities (n=266;73%), forensic inpatient facilities (n=50; 14%), long-term care/nursing homes (n=25; 7%), mixed inpatient settings (n=20; 5%), a learning disability unit (n=1; <1%), a health board (n=1; <1%) and a specialised research unit (n=1; <1%). More information about the study designs used is included in online supplementary file 4.

## Quality assessment
Most research was assessed as 'fair' quality (n=251; 69%), 86 (24%) papers were assessed as 'good' quality and 26 (7%) were assessed as 'poor' quality. None was assessed as 'very poor' quality. Studies rated as 'poor' mainly did not discuss ethical considerations, potential biases or give sample or setting characteristics. For example, they did not consider recruitment strategies, sample demographics or provide detailed information on the research setting. All 'good' studies provided setting and sampling information to allow for replicability. In addition, 'good' studies provided detail on data analysis justification, more thorough literature reviews to place the study in context and had clear research aims/objectives. Online supplementary file 5 includes a table showing the breakdown of the quality domain scores for each paper.

## Synthesis
Ten research categories were identified: interpersonal violence, coercive interventions, safety culture, harm to self, safety of the physical environment, medication safety, unauthorised leave, clinical decision making, falls and infection prevention and control. Within these categories 46 subcategories were identified (table 1).

## Interpersonal violence
Interpersonal violence was the largest category (n=116; 32%). Studies were primarily concerned with the prevalence, management and prevention of violent and aggressive behaviours (n=75). The pooled prevalence for physical violence was 43.2% (95% CI 0.37 to 0.49) with high heterogeneity ($I^2$ 100.0%) in 20 studies[20–39] (online supplementary file 6). The pooled prevalence for verbal aggression was 57.4% (95% CI 0.34 to 0.81) with a high heterogeneity ($I^2$ 100.0%) in 10 studies[22–24 26 29 33–36 40] (online supplementary file 6).

One study examined the characteristics of aggressive incidents by ward type,[41] and two studies identified correlates of violence.[42 43] One study explored how patients described their aggressive behaviours.[44] Twenty-four studies evaluated intervention effectiveness (eg, staff training and medication use) to reduce violent and aggressive behaviours, with most finding significant improvements,[45–65] two reporting negative outcomes[66 67] and one reporting mixed findings.[68] The general management of violent and aggressive behaviours was explored

**Table 1** Overview of study characteristic identified within each category

| Category | Subcategory | Category definition | Number of studies | Countries | Number of studies using staff participants | Number of studies using patient participants | Total number of participants | Settings (number of studies conducted in each setting) |
|---|---|---|---|---|---|---|---|---|
| Interpersonal violence | Aggression<br>Violence<br>Challenging behaviour<br>Violence and aggression<br>Critical incidents<br>Conflict<br>Sexual Assault<br>Agitation | Behaviours or events that are considered hostile with the intent to cause harm, including violence, aggression and conflicts. This also encompasses sudden emergency incidents that require management. | 115 | UK-31<br>USA-20<br>Australia-9<br>Canada-7<br>The -6<br>Sweden-7<br>Taiwan-4<br>South Africa-2<br>Switzerland-2<br>India-3<br>Italy-2<br>Turkey-3<br>Europe-2<br>New Zealand-1<br>South Korea-1<br>Finland-3<br>Greece-1<br>Spain-1<br>Hong Kong-1<br>Israel-2<br>Nigeria-1<br>Norway-2<br>Denmark-1<br>Japan-1<br>Germany-1<br>Slovakia-1 | 52<br>22 mixed | 39<br>22 mixed<br>1 family member of patients<br>1 N/A | 20066 (excl. missing data) | Psychiatric inpatient wards/facilities- 73<br>Forensic inpatient facilities-22<br>Long-term care/nursing homes-13<br>Specialised research unit-1<br>Mixed-6 |
| Coercive interventions | Restraint<br>Seclusion<br>Attitudes to coercion<br>Seclusion and restraint<br>Containment<br>Process of coercion<br>Alternative interventions<br>Shielding<br>Conflict<br>Personal factors | Techniques for managing patient behaviour that are applied without consent, for the safety of the patient and others. These include seclusion, restraint and containment. | 99 | UK-31<br>Finland-7<br>USA-8<br>The Netherlands-5<br>Australia-5<br>Canada-7<br>Norway-4<br>Germany-2<br>Sweden-3<br>Japan-2<br>Mixed-5<br>New Zealand-2<br>Europe-1<br>China-1<br>Switzerland-3<br>South Korea-1<br>India-2<br>Brazil-1<br>Denmark-1 | 36<br>29 mixed | 34<br>29 mixed | 59 732 (excl. missing data) | Psychiatric inpatient wards/facilities- 74<br>Forensic inpatient facilities-13<br>Long-term care/nursing homes-3<br>Mixed-8<br>Health board-1 |

Continued

**Table 1** Continued

| Category | Subcategory | Category definition | Number of studies | Countries | Number of studies using staff participants | Number of studies using patient participants | Total number of participants | Settings (number of studies conducted in each setting) |
|---|---|---|---|---|---|---|---|---|
| Safety culture | Process Culture Policy Building therapeutic relationships Patient/family engagement | The organisational attitudes, beliefs and values concerning safety. This encompasses the policies and procedures within the healthcare organisation in relation to safety. | 49 | UK-12 Australia-10 USA-5 Sweden-4 Finland-4 Canada-2 Ireland-2 The Netherlands-1 Greece-1 Italy-1 Germany-1 Belgium-2 Taiwan-1 Europe-1 Iran-2 | 33 13 mixed | 3 13 mixed | 59 420 (excl. missing data) | Psychiatric inpatient wards/facilities- 36 Forensic inpatient facilities-8 Long-term care/nursing homes-1 Mixed-4 |
| Harm to self | Self-harm Suicidal behaviour Self-neglect | The ways in which the healthcare system attempts to prevent, mitigate or manage deliberate behaviours displayed by patients that are intended to cause harm or death to themselves. | 36 | USA-11 UK-8 Ireland-3 Norway-4 The Netherlands-2 Sweden-2 Taiwan-2 Australia-1 Japan-2 Belgium-1 | 16 3 mixed | 17 3 mixed | 3631 (excl. missing data) | Psychiatric inpatient wards/facilities- 29 Forensic inpatient facilities-3 Long-term care/nursing homes-2 Learning disability homes-1 Mixed-1 |
| Safety of the physical environment | Security Environmental design Transitions of care Patient distribution Staffing Ligatures | The factors related to the physical environment of the healthcare setting that could impact on safety. This includes ligature points, staffing, security (door locking) and patient distribution. | 21 | UK-6 The Netherlands-3 USA-4 Australia-4 Germany-2 Mixed-1 Sweden-1 | 6 8 mixed | 7 8 mixed | 3140 (excl. missing data) | Psychiatric inpatient wards/facilities- 17 Forensic inpatient facilities-1 Long-term care/nursing homes-3 |

Continued

**Table 1** Continued

| Category | Subcategory | Category definition | Number of studies | Countries | Number of studies using staff participants | Number of studies using patient participants | Total number of participants | Settings (number of studies conducted in each setting) |
|---|---|---|---|---|---|---|---|---|
| Medication safety | Adverse events<br>Medication administration<br>Medication management<br>Medication dispensing<br>Adherence<br>Substance use | Mistakes made at any stage of the medication use process, from preparation, to administration and recording. This includes adverse drug events (or injuries that are the result of a drug-related intervention) and issues surrounding drug/alcohol use. | 17 | UK-7<br>Turkey-1<br>Spain-2<br>The Netherlands-1<br>Croatia-1<br>Germany-1<br>Denmark-1<br>Canada-2<br>Mixed-1 | 9<br>1 mixed | 7<br>1 mixed | 2396 (excl. missing data) | Psychiatric inpatient wards/facilities- 13<br>Forensic inpatient facilities-2<br>Long-term care/nursing homes-1<br>Mixed-1 |
| Unauthorised leave | Absconding<br>Wandering | The act of a patient leaving the healthcare setting without the knowledge or consent of staff/carers. This can be either with (absconding) or without intent (wandering) on the part of the patient. | 11 | UK-4<br>Australia-3<br>USA-1<br>Canada-1<br>Italy-1<br>Indonesia-1 | 3<br>1 mixed | 7<br>1 mixed | 978 (excl. missing data) | Psychiatric inpatient wards/facilities- 10<br>Long-term care/nursing homes-1 |
| Clinical decision making | Incident management<br>Risk assessment<br>Diagnosis | Incorrect diagnoses, risk assessments and other decision making processes of healthcare staff that impact on the safety of a patient. | 9 | USA-3<br>UK-3<br>Canada-1<br>Greece-1<br>The Netherlands-1 | 6 | 3 | 529 | Psychiatric inpatient wards/facilities- 8<br>Forensic inpatient facilities-1 |
| Falls | Falls<br>Injuries | Falling events that lead to the unintentional harm of an individual. This includes trips and injuries such as fractures. | 6 | USA-3<br>Sweden-2<br>Israel-1 | 3 | 3 | 180 (excl. missing data) | Psychiatric inpatient wards/facilities- 5<br>Long-term care/nursing homes-1 |
| Infection prevention and control | Infection prevention and control | Preventing harm caused by infection to patients and health workers. | 1 | Taiwan-1 | 1 | 0 | 13 | Psychiatric inpatient wards/facilities- 1 |

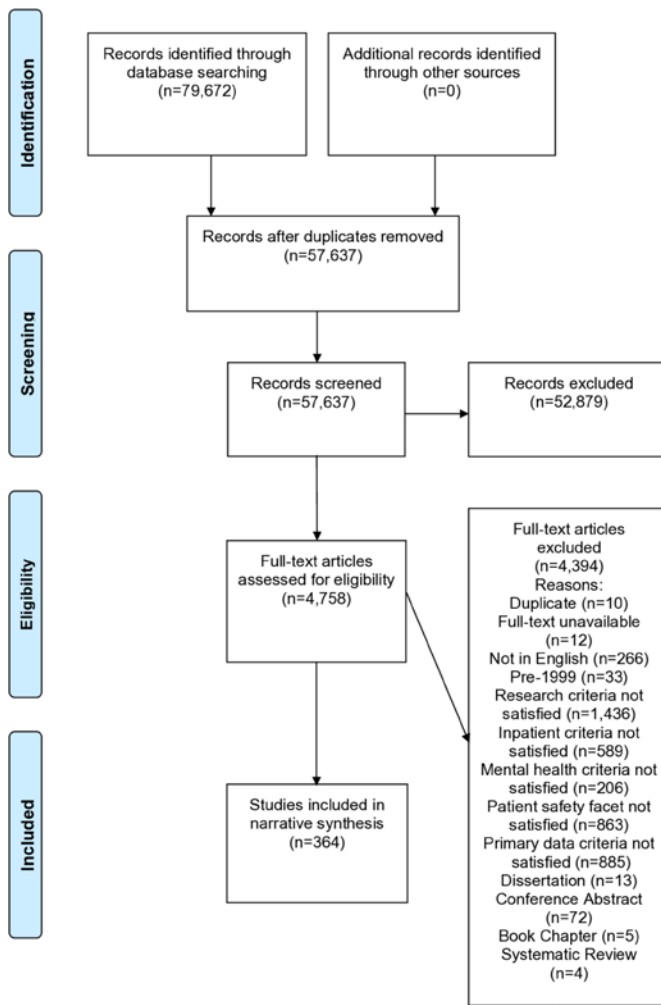

**Figure 1** Flow chart of studies.

in 15 studies.[22 25 29 30 69–79] Two studies explored the ways in which treatment can affect violence incidence.[31 80]

Twenty-seven studies explored violent and aggressive incident experiences in staff,[81–96] patients,[97–99] mixed groups[100–106] and patient family members.[27] Five studies explored the risk factors associated with verbal and physical aggression.[35 37 107–109] Three studies explored mental health nurses' perspectives on the response to violent situations in high secure environments: one on the psychological impact of physical assault on staff,[110] one on making violence risk assessments in imminent violent situations[111] and one on the decline of incident reports.[112] One study explored the link between aggressive behaviour and levels of burnout in staff[113] and one study looked at the role of social support for staff following a violent incident.[32]

Ten studies[114–123] examined challenging behaviour and techniques, such as de-escalation and communication strategies, which could be used to manage this; seven studies found techniques that were effective.[114–120] A further four studies investigated conflict behaviour management techniques employed by staff[124–126] and patients[127]; techniques used in the two intervention studies were effective in reducing conflict.[126 127] Staff and patient attitudes towards critical incidents were the focus

of four qualitative studies[128–131]; a further three studies focused on maintaining the psychological safety of patients who had experienced physical or sexual assault during an inpatient stay[132] and outside of healthcare.[133 134] Finally, one study explored an acupressure intervention to reduce agitation, which was found to be effective.[135]

### Coercive interventions

Coercive interventions were the focus of 98 papers (27%). Most studies (n=42) reported on restraint and seclusion techniques. The pooled prevalence for coercive interventions was 47.8% (95% CI 0.38 to 0.57) with high heterogeneity ($I^2$ 100.0%) in 12 studies[136–147] (online supplementary file 6).

Studies explored staff,[148–157] patient[147 158–165] and mixed groups'[166–173] views and experiences of seclusion and restraint. Nine studies focused on the processes surrounding seclusion and restraint.[136 137 174–180] A further 16 studies evaluated interventions to reduce seclusion and restraint, with 13 finding significant decreases in rates of use,[146 181–192] one reporting an increase[193] and one reporting increased levels of knowledge about the topic area.[194] Four studies examined prevalence, trends and preventative factors[138 195–197]; one found that 45% of patients were subjected to restraint,[138] and another found that restraint and seclusion declined over time.[197] One study explored the context in which seclusion and restraint had taken place.[198] Two studies found preventative factors of mechanical restraint to be staff education and increased patient involvement.[195 196] The training of staff in techniques for seclusion and restraint were explored in two studies[199 200] and one study examined adverse events resulting from restraint and seclusion.[201] Other studies explored staff and patient views of containment measures,[202–205] Maori views of initiatives to reduce/prevent seclusion,[206] the process of shielding (segregation under staff supervision),[207] conflict management[208] and alternative interventions.[209]

Thirty-two studies focused on coercion; one study examined prevalence of coercive measures[141] and one study explored how the experience of staff might contribute to the use of restrictive practices.[210] The attitudes of staff,[142 144 211–222] patients[145 223–226] and mixed groups[143 168 227–230] towards coercion were explored in 25 studies, and 5 studies examined the process of coercive interventions[139 140 231 232] and rules of engagement in caring for aggressive patients.[233]

### Safety culture

Safety culture included studies on processes, culture and policy across 49 papers (13%). Eighteen studies concerned safety-related organisational processes. Eleven of these investigated processes of treatment or care that healthcare staff undertake; processes included limit-setting and clothing restrictions,[234–240] risk assessment[241–243] and nursing handover.[244] Two investigated errors and reporting[245 246] and a further two studies explored staff and patient perceptions of safety when

involved in treatment processes.[247 248] Two studies focused on change implementation.[249 250] One study focused on the role of training.[251] Safety culture was featured in 18 publications relating to the management of serious incidents,[252–254] stress and burnout,[255–257] staff[258] and patient perspectives of safety[259–263] and communication[264]; there were also three papers that explored safety culture more generally.[265–267] A further two evaluated the TeamSTEPPS (Team Strategies and Tools to Enhance Performance and Patient Safety) programme[268 269] and both found significant clinical benefits in reducing seclusion and improving team functioning. One paper looked at the barriers and facilitators to implementing a Safewards intervention.[270] With regard to policy, eight studies concerned safety policies related to: observation,[271 272] risk assessment,[273 274] treatment,[275] safeguarding,[276] security[277] and ergonomic improvement.[278] Two papers focused on the role of patient and family engagement in safety,[279 280] and two papers focused on how to build better therapeutic relationships to improve patient safety.[281 282]

## Harm to self

Three subcategories centred on harmful behaviours: self-harm, suicidal behaviour and self-neglect (n=36; 10%). Half of the studies (n=18) focused on self-harm. One paper explored the prevalence of self-harm.[283] Two studies explored risk factors for self-harm which included use of psychotropic medication.[284 285] Eight papers explored staff attitudes and experiences of managing self-harm,[286–293] and three explored patient experiences.[294–296] Three intervention studies focused on training,[297] therapy[298] and observation[299]; all reported a reduction in self-harm behaviours and a further intervention focusing on training for staff resulted in positive attitude towards self-harm patients, greater closeness and improved self-efficacy.[300] Of the 17 papers that centred on suicidal behaviours, five studies investigated the observance of risk factors[301–305] and three intervention studies found significant reductions in suicide-related behaviours and cognitions.[306–308] An additional eight papers explored staff,[309–312] patient[313 314] and both staff and patient[315 316] views and attitudes towards suicidal behaviour. One study looked at the acceptability of an intervention to reduce suicide.[317] Finally, one study explored types of self-neglect behaviours in patients with dementia, including functional difficulties, serious hygiene problems and safety risks.[318]

## Safety of the physical environment

The safety of the physical environment category included 21 papers (6%). Seven studies investigated security measures (eg, door locking).[319–325] Five studies investigated the effects of the physical environmental design on the safety of treatment settings.[326–330] Three studies focused on safety during transitions of care,[331–333] with most based in dementia care settings. Three studies examined how the location of patients within the hospital setting can impact on safety, focusing on topics such as:

privacy, female-only wards and the use of segregated or combined wards/units.[334–336] The remaining three studies concerned staffing levels[337 338] and ligature points.[339]

## Medication safety

The medication safety category included 17 publications (5%). Five studies focused on adverse events, and examined: antipsychotics side effects,[340] how best to manage the effect of psychotropics on long QT segments,[341] antidepressants[342] and medication error reporting.[343 344] Three studies investigated errors occurring in broader medication management processes[345–347] and a further five studies focused on medication administration specifically.[348–352] The only intervention study aiming to reduce these errors found that a new medication dispensing system did not have any significant impact on patient safety.[353] Two studies explored staff perceptions of illicit substance use.[354 355] One further study described the development of a medication adherence intervention for patients who are prescribed mood-stabilising medication for bipolar disorder.[356]

## Unauthorised leave

Unauthorised leave included 11 publications (3%). Three explored the patient experience of absconding, specifically relating to patient perspectives of treatment and involuntary commitment.[357–359] One study explored staff perspectives of absconding management techniques,[360] and two studies evaluated interventions to reduce absconding rates; both were found to be effective.[361 362] Two studies focused on wandering behaviour in women with dementia, linking wandering to physical environment factors, such as light, sound, crowding[363] and falls.[364] The pooled prevalence of wandering behaviour was 50.2% (95% CI 0.49 to 0.52) with high heterogeneity ($I^2$ 78.0%) in two studies[363 364] (online supplementary file 6). The final three studies examined the consequences[365 366] and security measures surrounding absconding.[367]

## Clinical decision making

Clinical decision making accounted for 2% of the included publications (n=9). These publications covered the development of clinical judgements and decisions relating to incident management, risk assessment and diagnosis. Two studies explored the cultural differences considered by clinicians in the diagnosis of African-American patients.[368 369] Clinical decisions on whether to engage in seclusion and/or restraint were explored in five studies[370–374] and two studies explored the variation in assessment and prediction of violence between staff and settings.[375 376]

## Falls

Publications on falls formed the second smallest category within the review (n=6; 1%). Studies in this category focused on fall prevalence, falls experienced by older psychiatric inpatients with dementia and prevention/harm reduction techniques. A recurring risk factor for falling was found to be medication use.[377–379] Two fall

prevention intervention studies did not identify significant benefits,[380 381] and one study explored barriers and facilitators to such interventions.[382]

## Infection prevention and control

One paper (<1%) focused on staff experiences of infection prevention and control in psychiatric clinical settings.[383]

## DISCUSSION
### Main findings

This is the first review to examine patient safety within inpatient mental health settings that uses robust systematic methodology. As a result, we have identified ten research categories: interpersonal violence, coercive interventions, safety culture, harm to self, safety of the physical environment, medication safety, unauthorised leave, clinical decision making, falls and infection prevention and control. In addition, we have been able to include a meta-analysis of incidence and prevalence of aggression (verbal and physical), coercive intervention and wandering behaviour as well as providing an easily accessible reference index of literature in the inpatient mental health and patient safety domain. Previous reviews on this topic had focused on collating the literature on a restricted number (n=8) of predefined patient safety incidents (eg, violence and aggression),[7] or the concept of patient safety in inpatient mental health setting more broadly (eg, organisation management).[10] As such, the findings presented here offer a contemporary view of the breadth and depth of patient safety research in inpatient mental health settings.

We were concerned to see that only 364 papers were identified as a result of our comprehensive search. Although this can be seen as a large number of publications for a systematic review, it is a relatively small number to cover the care of a wide range of patients in a variety of inpatient mental health settings over a 20-year period (around 18 papers per year across all countries). While important work not meeting our inclusion criteria (eg, quality improvement initiatives and studies using secondary analysis of data) may have focused on patient safety in mental health, the lack of prospective peer-reviewed publications adds to the ongoing discussion surrounding the disparity in research focusing on patient safety in physical and mental healthcare.[384] In addition, there was a paucity of high-quality research in the area; just over two-thirds of the studies were considered to be 'fair', and only nine studies included in the meta-analysis were deemed 'good'. 'Poor' studies most frequently did not have clear research aims and objectives, study details were missing (eg, sample(s) and setting(s) used) and they failed to discuss issues related to ethical and researcher bias. Some qualitative studies explored both staff and patients' perspectives, an important aspect of research, particularly when safety in this context is a relatively new area of knowledge. However, there was limited intervention research,

particularly randomised controlled trials (RCTs). In the RCTs that were identified, sample sizes were mostly small.

The findings from the review also challenged our expectations in terms of breadth and depth of research. For example, we expected to find many publications on the prevention of suicide within inpatient settings due to the severity of harm. However, only one study that met inclusion criteria discussed suicide in relation to ligature points.[339] A scoping review also found only this one study, suggesting a consistency of approach.[385] This indicates that while the prevention of suicide is a well-established aspect of patient safety, it is now reviewed routinely, using pre-existing and secondary data, rather than through empirical research.

We also found little research focusing on the concepts required for system level reform,[386] which was disappointing seeing as some improvements have been made in physical healthcare.[387] For example, in line with research in the physical health domain,[388 389] we were hoping to find several studies exploring how patient and family engagement in care can promote patient safety.[390] However, only two studies identified in our review had patient/family engagement as their primary focus.[279 280] Similarly, we were expecting to identify literature investigating the lack of integration between physical and mental healthcare and the impact it has on patient safety.[391] However, the need to prevent and manage co-existing physical ill health was not identified in the review. This is surprising as patients with serious mental illness are twice as likely to die prematurely and much more likely to develop long-term conditions or become disabled, as those without serious mental illness.[392] This patient group is also vulnerable to asphyxiation during restraint and rapid tranquilisation.[393]

Research on medication safety in inpatient mental health settings was also limited in this review. This was unexpected considering two-thirds of patients with mental health problems are prescribed medication and are therefore potentially at risk of experiencing a medication safety incident. Research pertaining to falls was also limited, contrasting with patient safety research within the physical health domain that includes a focus on slips, trips and falls.[394]

### Strengths and limitations

We used a robust patient safety taxonomy to provide a comprehensive list of all incident types. This resulted in a wide coverage of publications in terms of setting, country and population. We systematically searched, screened, extracted and appraised data. As a result, our systematic review draws together all relevant literature concerning patient safety within inpatient mental health settings, simultaneously operating as an index resource for clinicians and researchers.

There were several limitations. We used the definition of patient safety given by Vincent[1] to guide this review. While this is more nuanced than the original Institute

of Medicine definition of safety 'freedom from accidental injury'[395] and is widely accepted within the patient safety movement, it may be that a more suitable definition reflects the specific challenges within the inpatient mental health setting.[396] This review only included peer-reviewed studies with primary data. Therefore, literature using secondary data such as pre-existing datasets and data from internal audits was excluded as it did not fulfil the criteria of being a prospective research study with clear research aims.[397] For example, data examined by the National Confidential Inquiry into Suicide and Homicide by People with Mental Illness is collected retrospectively from various sites across the country and would have been excluded from this review.[398] Moreover, non-peer-reviewed quality improvement reports have also been excluded. The decision was made to only include peer-previewed studies with primary data due to (i) the large number of potential publications in this area, (ii) the need to define the scope and focus of the review and (iii) the need for specificity as well as sensitivity. The investigation of patient safety in mental health inpatient settings using secondary data or in non-peer-reviewed formats is an avenue for additional systematic reviews.

The last systematic literature search was conducted on 27 June 2019, meaning that literature published since this date will not have been included. In order to further build on the review published here, a *living* systematic review (an ongoing updated summary of high-quality research)[399] would continue to identify relevant literature in this area. In terms of the meta-analysis, there was expected statistical and methodological variability in studies, particularly for physical and verbal aggression. It is possible that this was due to the inclusion of different definitions of aggression, time periods and type of inpatient setting. In relation to the agreement between reviewers (including the use of recommended piloting of inclusion and exclusion criteria within the screening stage),[400] inter-rater reliability calculations only achieved substantial agreement ($\kappa=0.61–0.80$) at both the title and full-text screening stages. Although higher kappas have been reported in other systematic reviews, a substantial agreement is classified as more than acceptable.[401]

While the research spanned five continents, the UK, the USA and Australia contributed over 50% of the included studies, leading to a potential cultural bias in the body of research identified within the review. We recommend that, where possible, future systematic reviews incorporate manuscripts in languages other than English to establish greater insight into the global literature on patient safety in inpatient mental health settings, with a view to limiting any cultural bias. Similarly, while the removal of publications denoting non-inpatient setting restricted the conclusions to the inpatient setting, issues pertaining to this environment are likely to be different to that of community, primary or social care settings. Additionally, studies were excluded before 1999 to coincide with the release of the Institute of Medicine's report 'To Err is Human: Building a Safer Health System'[395]; this may have narrowed the review scope as the historical context was minimised.

## Clinical implications and future research

This review informs academics, clinicians and service providers about the evidence base in the patient safety field within inpatient mental health settings. The findings allow researchers and clinicians to be directed to literature relevant to a given patient safety topic area, a useful starting point when developing practice guidelines.[402] Similarly, the findings may influence clinical practice, with those implementing interventions or designing service changes being able to easily access the current scientific understanding.

Future research should be informed by patient safety science more broadly and focus on filling the knowledge gaps highlighted in this review, that is, studies that explore (i) systems level improvement, (ii) patient and carer engagement in safety, (iii) suicide prevention across different countries, (iv) the nature of medication safety in inpatient mental health settings and (v) the prevalence and impact of staff to patient violence. These findings support our previous expert consensus study where academic and service user experts agreed that patient-driven research studies were needed.[403] The limited rigorous research surrounding patient safety within inpatient mental health settings necessitates future studies to: (i) include large inpatient samples relevant to the research design, (ii) perform appropriate intervention testing and (iii) examine safety from different perspectives. It should also focus on high-quality reporting of research, paying particular attention to the area of ethics, sampling and setting characteristics.

## CONCLUSION

This is the first systematic review to comprehensively examine research on patient safety within inpatient mental health settings. It has drawn together the existing literature and shed light on the gaps in knowledge. Inpatient mental health settings may demonstrate unique patient safety challenges and more research is needed to achieve parity with physical health. Addressing this through a strong body of evidence, informed by patient safety science more broadly, will mean that mental healthcare policy makers are in a better position to address safety issues, and implement robust and evidence-based interventions to improve care.

**Acknowledgements** The authors would like to thank the librarians at St Mary's Library, Imperial College London for their support with the study.

**Contributors** BT, LD, SCR, SAr and DD contributed to the design, data searches, data extraction, synthesis and writing of the report. HA contributed to the design, data extraction, meta-analysis and writing of the report. SAd and AD contributed to the design and synthesis, as well as writing and critically reviewing the report.

**Funding** This work is supported by the National Institute for Health Research (NIHR) Imperial Patient Safety Translation Research Centre. Infrastructure support was provided by the NIHR Imperial Biomedical Research Centre.

Competing interests  None declared.

Patient consent for publication  Not required.

Provenance and peer review  Not commissioned; externally peer reviewed.

Data availability statement  All data relevant to the study are included in the article or uploaded as supplementary information.

ORCID iDs
Lindsay Helen Dewa http://orcid.org/0000-0001-8359-8834
Hutan Ashrafian http://orcid.org/0000-0003-1668-0672
Stephanie Archer http://orcid.org/0000-0003-1349-7178

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
