## [Reviewer comments · BMJ Open]

ARTICLE DETAILS

TITLE (PROVISIONAL)	Patient safety in inpatient mental health settings: a systematic review
AUTHORS	Thibaut, Bethan; Dewa, Lindsay; Ramtale, Sonny; D'Lima, Danielle; Adam, Sheila; Ashrafian, Hutan; Darzi, Ara; Archer, Stephanie

VERSION 1 - REVIEW

REVIEWER	Kathleen Delaney Rush College of Nursing
REVIEW RETURNED	10-Apr-2019

GENERAL COMMENTS	Thank you for the opportunity to review this quite amazing paper. A large portion of inpatient psychiatric nurses' attention is directed to safety and maintaining a safe environment. This paper provides a succinct summary of where we are in these efforts. I found the nine categories logical and consistent with the data. Table one was very helpful in diving a bit deeper into the categories. I agree with the authors' assertion that this review will serve as an excellent starting off point for any investigator aiming to research a particular area. It is also a good progress report and "wake up" call for those of us who have conducted research in the field. Areas that left me with questions are few. They are: 1. On the incidence of violence- I am assuming that the patient and staff were combined as the victim of physical aggression so you do not include injury in restraining incidents?2. On medication safety – or maybe clinical judgement- I believe an emerging area is nurses and appropriate PRN use (particularly benzodiazepines)- might want to include in comments on emerging issues.3. Another emerging topic is the idea of engagement. I think engagement begins to tie into safety culture – but we need to get more explicit to make that connection.4. Just a comment, being from the US, we just saw a report on the estimated amount/costs of environmental re-design around ligatures, interesting that you found one study on this. Thanks for the work.
---

REVIEWER	Morgan C. Shields Brandeis University, United States
REVIEW RETURNED	18-Apr-2019

GENERAL COMMENTS	I thank the authors for their work in compiling a review of patient safety within inpatient psychiatric care, which is a neglected area of research compared to other areas of medicine and where patients are particularly vulnerable. This paper will make a contribution to the literature given responsiveness to the following comments. In summary, there are some minor comments in addition to more major ones. Regarding the major comments, I suggest the authors take another pass through the literature, dropping the requirement for primary data collection and potentially including years 2017 and 2018. Abstract: I suggest the authors highlight the importance of the review in the objective. They rightly describe the importance of looking at patient safety and how (in one way) it is different from other areas of medicine, but do not mention the need for a synthesis and how this paper fills that gap in the literature. The authors state this in the first sentence of the Article Summary. Introduction: I suggest the authors call out the source for their definition of patient safety. Since they use the IOM report as a cutoff point for publication date, why do they not use the IOM definition of safety? However, the definition of safety in other areas of medicine might not be completely applicable to inpatient psychiatry. Methods: It appears the authors used a much more exhaustive set of search terms than what they mention in their methods, which is confusing because when reading the methods. I didn't realize how extensive their search was until I looked at the "supplemental 2" document. To prevent readers from assuming the authors were not rigorous with their search, I suggest the authors try to communicate this better in the methods. I question if the authors should update their search to include papers published in 2017 and 2018, since it is already 2019 and this paper potentially will not be published until 2020 --- making 2016 already dated. However, I will defer to the editor on this. I suggest eliminating the exclusion restriction of secondary data and updating the search. It is unclear to me why secondary data was an exclusion condition. A great deal of high-quality/value research on patient safety leverages secondary data analysis, such as retrospective reviews of patients' medical charts for identification of adverse events and medical errors. Some of the largest studies on harms to patients within inpatient psychiatry were excluded from the authors' review and this is concerning. Just a few are: Grasso, B. C., Genest, R., Jordan, C. W., & Bates, D. W. (2003). Use of chart and record reviews to detect medication errors in a state psychiatric hospital. Psychiatric services, 54(5), 677-681.
--

Hanrahan, N. P., Kumar, A., & Aiken, L. H. (2010). Adverse events associated with organizational factors of general hospital inpatient psychiatric care environments. *Psychiatric Services*, 61(6), 569-574

Maidment, I. D., Lelliott, P., & Paton, C. (2006). Medication errors in mental healthcare: a systematic review. *BMJ Quality & Safety*, 15(6), 409-413.

(these are from 2017, but I think the authors should do an update to include 2017):

Vermeulen, J. M., Doedens, P., Cullen, S. W., van Tricht, M. J., Hermann, R., Frankel, M., ... & Marcus, S. C. (2018). Predictors of adverse events and medical errors among adult inpatients of psychiatric units of acute care general hospitals. *Psychiatric Services*, 69(10), 1087-1094.

Marcus, S. C., Hermann, R. C., Frankel, M. R., & Cullen, S. W. (2017). Safety of psychiatric inpatients at the Veterans Health Administration. *Psychiatric services*, 69(2), 204-210.

Alshehri, G. H., Keers, R. N., & Ashcroft, D. M. (2017). Frequency and nature of medication errors and adverse drug events in mental health hospitals: a systematic review. *Drug safety*, 40(10), 871-886.

Goulet, M. H., Larue, C., & Dumais, A. (2017). Evaluation of seclusion and restraint reduction programs in mental health: A systematic review. *Aggression and violent behavior*, 34, 139-146.

Also excluded from the review are papers that evaluated emotional and psychological harm, including hospital-acquired post-traumatic stress (which did use primary data collection):

Frueh, B. C., Knapp, R. G., Cusack, K. J., Grubaugh, A. L., Sauvageot, J. A., Cousins, V. C., ... & Hiers, T. G. (2005). Special section on seclusion and restraint: Patients' reports of traumatic or harmful experiences within the psychiatric setting. *Psychiatric Services*, 56(9), 1123-1133.

Mayers, P., Keet, N., Winkler, G., & Flisher, A. J. (2010). Mental health service users' perceptions and experiences of sedation, seclusion and restraint. *International Journal of Social Psychiatry*, 56(1), 60-73.

Robins, C. S., Sauvageot, J. A., Cusack, K. J., Suffoletta-Maierle, S., & Frueh, B. C. (2005). Special section on seclusion and restraint: Consumers' perceptions of negative experiences and "Sanctuary Harm" in psychiatric settings. *Psychiatric Services*, 56(9), 1134-1138.

I suggest papers focused on nursing homes be excluded since the focus of this review is on the setting of inpatient psychiatric care and not psychiatric patients regardless of the setting. There is an entire literature focused on safety within nursing homes (where indeed psych patients can be but the focus here is on the setting of inpatient psych). Related to this, the authors' definition of

	“inpatient settings” should be specific to psychiatric/mental health inpatient settings. Discussion: I suggest the authors mention systems-level and organizational predictors (e.g., the role of profit-oriented organizations, incentives, data systems), and how there is dearth of systems-level research looking at patient safety. The focus is at the organizational level. Where the authors mention how their areas of research extend those of previous reviews (in the first paragraph), please state in which ways. What areas were identified in the previous reviews and how does this compare/add? I thank the authors again for this much needed and thoughtful review of the literature.
--	---

REVIEWER	Lisa M Brophy La Trobe University Australia
REVIEW RETURNED	13-May-2019

GENERAL COMMENTS	Congratulations on this excellent paper that makes a valuable contribution to the literature. I must admit that I was most frustrated by the limits of your findings - not through any problem with your design and method - but in the overall finding about the paucity of literature on patient safety in inpatient mental health settings. I was particularly surprised by the few papers you found regarding sexual safety. I am aware of a project conducted in Melbourne Australia that included a literature review https://www.mhcc.vic.gov.au/news-and-events/news/ensuring-sexual-safety-in-acute-mental-health-inpatient-units but on reviewing the reference list I can see that many are not included in your review because most were from the grey literature and they included peer reviewed publications prior to 1999. Also many did not exclusively focus on inpatient units. However, I think there has been considerable work done in this area since April 2016 so I hope your review is updated shortly. I also agree with your finding that the lack of empirical research regarding suicide prevention on inpatient units is challenging and problematic. I also agree that your review has been limited because of the emphasis on papers in English - leading to a cultural bias. However, I am satisfied that you have been clear in letting readers know about the limitations of your study. There are some specific minor revisions required to the manuscript:  1. I think it would be best to find an alternative to "most unwell" in the opening paragraph. This may not be universally understood or agreed with as a way to describe people on inpatient units. I suggest "experiencing high levels of mental distress" 2. Using Cohen's Kappa to calculate agreement rates requires a reference- I note that this is not mentioned in your protocol paper. It is also not clear how often consultation with a third reviewer was required. 3. In relation to quality assessment - it may be presumed by the authors but I think it is valuable here to state the purpose of the
---

	quality assessment for your review - especially since it did not lead to any papers being excluded. 4. Page 9 - "maintaining the patient psychological safety who had undergone sexual assault" seems to need rephrasing to clarify the meaning. Also "abused female patients" - what abuse does this refer to? In childhood? On inpatient units? 5. Page 12 -Accidents - this needs re phrasing - do you mean the "prevalence of falls" and could words be deleted here? 6. The paper is silent on the issue of staff representing a risk to patients in relation to assaults and abuse - I presume again that this reflects the paucity of literature but to many consumers this is an important although seemingly under investigated issue. You may consider adding this to you list of knowledge gaps
--	--

VERSION 1 – AUTHOR RESPONSE

Reviewer 1

Thank you for the opportunity to review this quite amazing paper. A large portion of inpatient psychiatric nurses' attention is directed to safety and maintaining a safe environment. This paper provides a succinct summary of where we are in these efforts.

Thank you for this encouraging feedback.

I found the nine categories logical and consistent with the data. Table one was very helpful in diving a bit deeper into the categories. I agree with the authors' assertion that this review will serve as an excellent starting off point for any investigator aiming to research a particular area. It is also a good progress report and "wake up" call for those of us who have conducted research in the field.

We are very pleased to hear that the paper will be useful to researchers and clinicians with an interest in the area of mental health and patient safety as well as the wider readership of BMJ Open.

Areas that left me with questions are few. They are:

On the incidence of violence- I am assuming that the patient and staff were combined as the victim of physical aggression so you do not include injury in restraining incidents?

Thank you for raising this – any papers that focused on injuries arising from restraining would have been included in the coercive intervention category in order to draw together all of the literature focusing on the incidence, impact and improvement of coercive practices. None of the papers included in this review had the incidence of injury resulting from restraining incidents as their main focus.

On medication safety – or maybe clinical judgement- I believe an emerging area is nurses and appropriate PRN use (particularly benzodiazepines)- might want to include in comments on emerging issues.

Thank you for raising this important point – we have included medication safety as an area ripe for future research on page 16. However, due to the limited word count, we do not feel able to be more specific on the types or topics of research within each of these areas.

Another emerging topic is the idea of engagement. I think engagement begins to tie into safety culture – but we need to get more explicit to make that connection.

Thanks for raising this point – we have included a note about the lack of research on patient/family engagement in a new section on system level reform on page 14 – the section on patient/family engagement reads:

For example, in line with research in the physical health domain, we were hoping to find studies exploring how patient and family engagement in care can promote patient safety. However, no studies identified in our review had patient/family engagement as their primary focus.

We have also included patient and carer engagement in safety in the list of future research areas; it has replaced patient and carer perspectives as we agree that the term engagement is more appropriate. The section on page 16 reads:

Future research should be informed by patient safety science more broadly and focus on filling the knowledge gaps highlighted in this review i.e. studies that explore (i) systems level improvement (ii) patient and carer engagement in safety, (iii) suicide prevention across different countries, (iv) the nature of medication safety in inpatient mental health settings and (v) the prevalence and impact of staff to patient violence.

Just a comment, being from the US, we just saw a report on the estimated amount/costs of environmental re-design around ligatures, interesting that you found one study on this.

Reviewer: 2

I thank the authors for their work in compiling a review of patient safety within inpatient psychiatric care, which is a neglected area of research compared to other areas of medicine and where patients are particularly vulnerable. This paper will make a contribution to the literature given responsiveness to the following comments.

Thank you for the positive feedback and we are glad to hear that you think the paper will be valuable to researchers and clinicians with an interest in mental health and patient safety.

In summary, there are some minor comments in addition to more major ones. Regarding the major comments, I suggest the authors take another pass through the literature, dropping the requirement for primary data collection and potentially including years 2017 and 2018.

Thank you for this feedback – we have addressed each of these points in more detail below.

Abstract:

I suggest the authors highlight the importance of the review in the objective. They rightly describe the importance of looking at patient safety and how (in one way) it is different from other areas of medicine, but do not mention the need for a synthesis and how this paper fills that gap in the literature. The authors state this in the first sentence of the Article Summary.

Thank you for this helpful suggestion. We have now included the following sentence on page 1:

The objective of this review is to identify and synthesise the literature on patient safety within inpatient mental health settings using robust systematic methodology.

Introduction:

I suggest the authors call out the source for their definition of patient safety. Since they use the IOM report as a cut-off point for publication date, why do they not use the IOM definition of safety? However, the definition of safety in other areas of medicine might not be completely applicable to inpatient psychiatry.

Thank you for this feedback. We include the definition from Vincent (2006) as it is widely accepted in more recent patient safety literature. However, we have noted in the limitations section (page 15) that alternative definitions may better reflect the challenges of the inpatient mental health setting:

We used the definition of patient safety given by Vincent to guide this review. Whilst this is more nuanced than the original Institute of Medicine definition of safety “freedom from accidental injury” and is widely accepted within the patient safety movement, it may be that a more suitable definition reflects the specific challenges within the inpatient mental health setting.

Methods:

It appears the authors used a much more exhaustive set of search terms than what they mention in their methods, which is confusing because when reading the methods. I didn't realize how extensive their search was until I looked at the “supplemental 2” document. To prevent readers from assuming the authors were not rigorous with their search, I suggest the authors try to communicate this better in the methods.

Thank you for this feedback – we have now included the number of terms included in each facet on page 4 to give some indication of the extent of the search. The text now reads”

The search was originally conducted on the 5th April 2016 and then updated on the 27th of June 2019 using a comprehensive list of search terms (n=343) related to “mental health” (n=73), “patient safety” (n=206), “inpatient setting” (n=13) and “research” (n=51); see online supplement 1 and 2 for full search criteria and terms.

I question if the authors should update their search to include papers published in 2017 and 2018, since it is already 2019 and this paper potentially will not be published until 2020 --- making 2016 already dated. However, I will defer to the editor on this.

Thank you for this suggestion. We have updated the search to 27th June 2019 – This process has identified an additional 94 papers. The process of identifying these papers and the information extracted from them has been included in the main body of the manuscript (Note: for the sake of brevity, we have not included all of the edited aspects in this document)

I suggest eliminating the exclusion restriction of secondary data and updating the search. It is unclear to me why secondary data was an exclusion condition. A great deal of high-quality/value research on patient safety leverages secondary data analysis, such as retrospective reviews of patients’ medical charts for identification of adverse events and medical errors. Some of the largest studies on harms to patients within inpatient psychiatry were excluded from the authors’ review and this is concerning. Just a few are [lists 7 papers].

Thank you for highlighting this interesting literature on patient safety on inpatient mental health settings. We agree that a number of insightful papers may not have been included in the review due to the exclusion of secondary data. The decision to exclude literature focusing on secondary analysis of data was something we debated at length when scoping our review. The decision to exclude these papers was based on three factors: (i) we wanted identify studies that prospectively set out to study patient safety in order to assess the state of research actively being conducted in this area (ii) we needed to provide strict include and exclude criteria which was difficult to achieve with the breadth of literature using secondary data and (iii) we needed to keep the review manageable for completion within our small team in a specific timeframe. Considering these three factors, we decided that it would be most effective to conduct a contained and manageable review that would be a useful starting point for further research in this area.

In order to highlight this, we had previously discussed the importance of literature that uses secondary data in the limitations section of the paper on page 19.

This review only included peer reviewed studies with primary data. Therefore, literature utilising secondary data such as pre-existing datasets and data from internal audits was excluded as it did not fulfil the criteria of being a prospective research study with clear research aims 293. For example, data examined by the National Confidential Inquiry into Suicide and Homicide by People with Mental Illness (NCISH) is collected retrospectively from various sites across the country 294 and would have been excluded from this review. Moreover, non-peer reviewed quality improvement reports would also

have been excluded as there was a need to define the scope and focus, due to the large number of potential publications in this area and the need for specificity as well as sensitivity. The investigation of patient safety in mental health inpatient settings using secondary data or in non-peer reviewed formats is an avenue for additional systematic reviews.

Following your feedback, we have also added an additional section to the discussion on page 14 which reads:

Whilst important work not meeting our inclusion criteria (e.g. quality improvement initiatives and studies using secondary analysis of data) may have focused on patient safety in mental health, the lack of prospective peer reviewed publications adds to the ongoing discussion surrounding the disparity in research focusing on patient safety in physical and mental healthcare¹¹.

Also excluded from the review are papers that evaluated emotional and psychological harm, including hospital-acquired post-traumatic stress (which did use primary data collection):

- Frueh, B. C., Knapp, R. G., Cusack, K. J., Grubaugh, A. L., Sauvageot, J. A., Cousins, V. C., ... & Hiers, T. G. (2005). Special section on seclusion and restraint: Patients' reports of traumatic or harmful experiences within the psychiatric setting. *Psychiatric Services*, 56(9), 1123-1133.
- Mayers, P., Keet, N., Winkler, G., & Flisher, A. J. (2010). Mental health service users' perceptions and experiences of sedation, seclusion and restraint. *International Journal of Social Psychiatry*, 56(1), 60-73.
- Robins, C. S., Sauvageot, J. A., Cusack, K. J., Suffoletta-Maierle, S., & Frueh, B. C. (2005). Special section on seclusion and restraint: Consumers' perceptions of negative experiences and "Sanctuary Harm" in psychiatric settings. *Psychiatric Services*, 56(9), 1134-1138.

Thank you for bringing this literature to our attention – unfortunately, these papers would not have met our 'inpatient' inclusion criteria because the paper did not explicitly specify that participants were inpatients (Mayers et al, 2010) or stated that the studies were conducted in 'day hospitals' (Frueh et al, 2005; Robin et al., 2005).

I suggest papers focused on nursing homes be excluded since the focus of this review is on the setting of inpatient psychiatric care and not psychiatric patients regardless of the setting. There is an entire literature focused on safety within nursing homes (where indeed psych patients can be but the focus here is on the setting of inpatient psych). Related to this, the authors' definition of "inpatient settings" should be specific to psychiatric/mental health inpatient settings.

Thank you for raising this interesting discussion point; again, the scope of 'inpatient' is something that we debated at length during the scoping phase of our review. Our 'inpatient' criteria was purposefully broad so to include studies conducted in environments where continuous healthcare was provided for a period of 24 hours or more. This was applied in conjunction with criteria relating to type of setting. We only included research undertaken in purely mental health settings. This meant that research from nursing homes would only have been included in the review if focused specifically on a mental health setting within the home (for example, a dementia unit embedded within a more general care home),

or if the home itself was designated a mental health setting (for example, specialised learning disability units).

Discussion: I suggest the authors mention systems-level and organizational predictors (e.g., the role of profit-oriented organizations, incentives, data systems), and how there is dearth of systems-level research looking at patient safety. The focus is at the organizational level.

Thank you for this feedback – we have now included a short section on the lack of papers focusing on system level reform on page 14. The text now reads:

We also found little research focusing on the concepts required for system level reform [381], which was disappointing seeing as some improvements have been made in physical healthcare [382].

We have also included systems level research in the list of future research areas on page 16:

Future research should be informed by patient safety science more broadly and focus on filling the knowledge gaps highlighted in this review i.e. studies that explore (i) systems level improvement (ii) patient and carer engagement in safety, (iii) suicide prevention across different countries, (iv) the nature of medication safety in inpatient mental health settings and (v) the prevalence and impact of staff to patient violence.

Where the authors mention how their areas of research extend those of previous reviews (in the first paragraph), please state in which ways. What areas were identified in the previous reviews and how does this compare/add?

Thank you for making this important point; we have removed reference to extending the findings and have better described the differences between our study and the previous reviews. We have added some additional text to speak to this point on page 13:

Previous reviews on this topic had focused on collating the literature on a restricted number (n=8) of pre-defined patient safety incidents (e.g. violence and aggression) [7], or the concept of patient safety in inpatient mental health setting more broadly (e.g. organisation management) [10]. As such, the findings presented here offer a contemporary view of the breadth and depth of patient safety research in inpatient mental health settings.

I thank the authors again for this much needed and thoughtful review of the literature.

Thank you for your helpful and thought-provoking review.

Reviewer: 3

Congratulations on this excellent paper that makes a valuable contribution to the literature.

I must admit that I was most frustrated by the limits of your findings - not through any problem with your design and method - but in the overall finding about the paucity of literature on patient safety in inpatient mental health settings. I was particularly surprised by the few papers you found regarding sexual safety. I am aware of a project conducted in Melbourne Australia that included a literature review <https://www.mhcc.vic.gov.au/news-and-events/news/ensuring-sexual-safety-in-acute-mental-health-inpatient-units> but on reviewing the reference list I can see that many are not included in your review because most were from the grey literature and they included peer reviewed publications prior to 1999. Also many did not exclusively focus on inpatient units. However, I think there has been considerable work done in this area since April 2016 so I hope your review is updated shortly. I also agree with your finding that the lack of empirical research regarding suicide prevention on inpatient units is challenging and problematic. I also agree that your review has been limited because of the emphasis on papers in English - leading to a cultural bias. However, I am satisfied that you have been clear in letting readers know about the limitations of your study.

Thank you for your thoughtful review of our paper. We were very surprised, and often disappointed, by some of the findings. We are glad to hear that our findings were consistent with your knowledge of the area. In order to bring the review up to date, we have updated the search to include papers from 1999 to 27th June 2019.

There are some specific minor revisions required to the manuscript:

I think it would be best to find an alternative to "most unwell" in the opening paragraph. This may not be universally understood or agreed with as a way to describe people on inpatient units. I suggest "experiencing high levels of mental distress".

Thank you for this useful feedback – we have edited the sentence on page 4 to read:

Inpatient services will often include patients who are experiencing high levels of mental distress and are therefore at greatest risk.

Using Cohen's Kappa to calculate agreement rates requires a reference- I note that this is not mentioned in your protocol paper. It is also not clear how often consultation with a third reviewer was required.

Thank you for highlighting these two omissions from our paper. We have now included the original Cohen (1960) reference and two others (McHugh, 2012 & Belur et al., 2018) that support its use.

Agreement between reviewers was calculated using Cohen's kappa [14], which is a widely accepted measure of interrater reliability [15, 16].

We have also included the number of times that the third reviewer was consulted. The text on page 7 reads:

A total of 4,758 publications were subjected to full-text review (BT, CR, LD and SAr). Two reviewers independently screened the full text articles against inclusion criteria (BT, CR, LD, and SAr). The third reviewer (DD) was consulted 59 times.

In relation to quality assessment - it may be presumed by the authors but I think it is valuable here to state the purpose of the quality assessment for your review - especially since it did not lead to any papers being excluded.

Thank you for providing this feedback – we have included a simple statement at the start of the quality assessment section on page 6 which reads:

Quality assessment was performed to give an overview of the methodological rigour of included studies and to support readers' interpretation of the literature. Publications were not excluded based on poor quality because the review was purposively exploratory and all-encompassing.

Page 9 - "maintaining the patient psychological safety who had undergone sexual assault" seems to need rephrasing to clarify the meaning. Also "abused female patients" - what abuse does this refer to? In childhood? On inpatient units?

Thank you for highlighting these areas that would benefit from additional clarity – these have now been amended on page 9 of the manuscript. The relevant sections now read:

A further three studies focused on maintaining the psychological safety of patients who had experienced sexual assault during an inpatient stay [130] and outside of healthcare [131, 132].

Accidents - this needs re phrasing - do you mean the "prevalence of falls" and could words be deleted here?

Thank you for highlighting this. We have now changed the category name to 'Falls', and have updated the manuscript throughout.

The paper is silent on the issue of staff representing a risk to patients in relation to assaults and abuse - I presume again that this reflects the paucity of literature but to many consumers this is an important although seemingly under investigated issue. You may consider adding this to you list of knowledge gaps

Thank you for highlighting this important area of future research. Two papers were identified in the updated search, but this is still fewer than we expected. As such, we have now included this in our list of topics ripe for future research on page 16:

Future research should be informed by patient safety science more broadly and focus on filling the knowledge gaps highlighted in this review i.e. studies that explore (i) systems level improvement (ii) patient and carer engagement in safety, (iii) suicide prevention across different countries, (iv) the

nature of medication safety in inpatient mental health settings and (v) the prevalence and impact of staff to patient violence.

VERSION 2 – REVIEW

REVIEWER	Morgan Shields Brandeis University
REVIEW RETURNED	01-Nov-2019

GENERAL COMMENTS	Thank you for this excellent contribution to the literature.
--

REVIEWER	Lisa Brophy La Trobe University
REVIEW RETURNED	10-Nov-2019

GENERAL COMMENTS	Well done on responding so respectfully and comprehensively to all three reviewers comments on your first submission. I have nothing further to add except that I really appreciate the efforts that must have been required to update the review - I think it is worth it.
---